# A protocol for evaluating the entomological impact of larval source reduction on mosquito vectors at hotel compounds in Zanzibar

**Ayubo Kampango**[1,2]*, **Fatma Saleh**[3], **Peter Furu**[4], **Flemming Konradsen**[4], **Michael Alifrangis**[5,6], **Karin L. Schiøler**[4], **Christopher W. Weldon**[2]

**1** Sector de Estudos de Vectores, Instituto Nacional de Saúde (INS), Vila de Marracuene, Província de Maputo, Mozambique, **2** Department of Zoology and Entomology, University of Pretoria (UP), Pretoria, South Africa, **3** Department of Allied Health Sciences, School of Health and Medical Sciences, The State University of Zanzibar, Zanzibar, Tanzania, **4** Global Health Section, Department of Public Health, University of Copenhagen, Copenhagen, Denmark, **5** Center for Medical Parasitology, Department of Immunology and Microbiology, University of Copenhagen, Copenhagen, Denmark, **6** Department of Infectious Diseases, Copenhagen University Hospital (Rigshospitalet), Copenhagen, Denmark

\* akampango@gmail.com

**Data Availability Statement:** The underlying data will be shared via the Open Science Framework (OSF) repository after the study has been

## Abstract

There is an increasing awareness of the association between tourism activity and risks of emerging mosquito-borne diseases (MBDs) worldwide. In previous studies we showed that hotels in Zanzibar may play an important role in maintaining residual foci of mosquito vectors populations of public health concern. These findings indicated larval sources removal (LSR) interventions may have a significant negative impact on vector communities. However, a thorough analysis of the response vector species to potential LSM strategies must be evaluated prior to implementation of a large-scale area-wide control campaign. Here we propose a protocol for evaluation of the impact of LSR against mosquito vectors at hotel settings in Zanzibar. This protocol is set to determine the efficacy of LSR in a randomized control partial cross-over experimental design with four hotel compounds representing the unit of randomization for allocation of interventions. However, the protocol can be applied to evaluate the impact of LRS in more than four sites. Proposed interventions are active removal of disposed containers, and installation of water dispenser to replace single use discarded plastic water bottles, which were identified as the most important source of mosquitoes studied hotels. The ideal time for allocating intervention to the intervention arms the dry season, when the mosquito abundance is predictably lower. The possible impact of interventions on mosquito occurrence and abundance risks is then evaluated throughout subsequent rainy and dry seasons. If an appreciable reduction in mosquito abundance and occurrence risks is observed during the trial period, intervention could be extended to the control arm to determine whether any potential reduction of mosquito density is reproducible. A rigorous evaluation of the proposed LRS interventions will inspire large scale trials and provide support for evidence-based mosquito management at hotel facilities in Zanzibar and similar settings.

implemented. The data will be available under the terms of the Creative Commons Zero "No rights reserved" data waiver (CC0 1.0 Public domain dedication).

**Funding:** Preparation of this material has been supported by external fundings from Denmark's Development Corporation (Danida), through the Danida Fellowship Centre (DFC), as part of the EnSuZa project Grant 17-04-KU. Additional internal fundings were obtained from the Danida's Building Stronger Universities (BSU) Phase III programme at the State University of Zanzibar (SUZA). There was no additional external funding received for this study.

**Competing interests:** The authors have declared that no competing interests exist.

## Introduction

Mosquitoes are vectors of several important parasitic and viral diseases that affect human health and halt economic and societal progress in endemic countries [1]. Malaria, dengue, chikungunya, yellow fever, Zika, Japanese encephalitis and lymphatic filariasis are the most notorious mosquito-borne diseases (MBDs) [2]. They are present across tropical and subtropical countries, with more than half of the world's population currently at risk [2].

Tanzania is one of the four countries accounting for half of the globally reported malaria deaths [3]. Notably, the country is also prone to large-scale outbreaks of mosquito-borne viral disease (MBVD). This includes *Aedes*-borne diseases, as evidenced by an unprecedented number of dengue virus outbreaks since 2010 [4–6]. The most recent outbreak occurred on the mainland in 2019, resulting in 6,670 confirmed cases including 13 deaths [7]. Remarkably, in the Zanzibar archipelago, dengue or related MBVD outbreaks have not been officially reported since 1823 [8]. However, there is a high risk of large-scale epidemics given widespread infestation by *Aedes* species [9–11], extensive connectivity between Zanzibar, mainland Tanzania and other endemic regions [12, 13], rapid demographic expansion [14], as well as poor solid waste management [15], all of which may precipitate the emergence of outbreaks in Zanzibar. Local studies and clinical reports of visitors returning from Zanzibar strongly suggest that there may be silent circulation of dengue and chikungunya virus in the archipelago [16–18]. This underscores the need to identify and eliminate the drivers of potential *Aedes*-borne disease epidemics in the region.

The impact of tourism on mosquito-borne disease expansion, and the increased risk of MBVD outbreaks due to tourism is gaining long-overdue global attention [19]. Tourism promotes connectivity between different sites, increasing the likelihood of introducing novel pathogens into new environments and exposing immunologically naïve visitors to potentially deadly pathogens at endemic travel destinations. Therefore, it is crucial for popular tourist destinations, such as Zanzibar to implement actions that would reduce the risk of epidemics and from becoming the source of international spread. This is vital, given the limited readiness of the Zanzibar healthcare system for management and control of mosquito-borne viral diseases [20]. In previous studies [9, 10, 21], we have shown that hotels in Zanzibar may provide suitable conditions for sustaining mosquito vectors of global concern, notably *Ae. aegypti*, *Ae. bromeliae*, *Ae. africanus*, *Ae. metallicus*, *Cx. quinquefasciatus*, *Er. quinqueviattatus*, *Er. chrysogaster* and, to less extent, *An. gambiae* [9, 10]. These vectors utilize discarded water-holding containers and domestic utensils as larval habitats, mainly plastic water bottles, plant pots, cans, and concrete water tanks [9, 10]. Occurrence of such vectors aligns with increasing visitor reports of mosquito bites during their stay at hotels in Zanzibar. Moreover, the fear of being infected by life-threatening mosquito-borne diseases may be influence travel destination choice, resulting in low interest in return to the region. Some visitors may not even consider Zanzibar in the first place, as the apparent risk of disease exposure governs the choice of tourism destination [19]. The tourism industry would be severely affected, in case of epidemic events, following a devasting impact on the economic stability of Zanzibar, as observed in several other insular regions strongly reliant on tourism [22–25]. Furthermore, we have also shown that resistance to DDT and pyrethroid insecticides is already established in the arbovirus vector species *Ae. aegypti* found at hotel compounds on the southeast coastal region of Zanzibar Island [21]. These findings strongly suggest that the studied hotels qualify for environmental mosquito management interventions, by means of larval source management (LSM), to control mosquito populations [26].

The management of nuisance as well as vector mosquitoes at hotel compounds has long relied on insecticide-based measures, notably outdoor space spraying/fogging and use of bed

nets [21]. Residual spraying and bed nets have proven inefficacious against important day-biting, mosquito vectors, such as *Ae. aegypti*, in several contexts [27], possibly due to a remarkable and widespread level of insecticide resistance [21, 28]. Wasteful use of inefficacious insecticides at hotel compounds can exacerbate the resistance selection problem and cause irreparable environmental harm in the form of biodiversity loss of non-target insect and fauna.

Several randomized trials have shown that interventions combining community-based mosquito larval source reduction (LSR) with consistent surveillance can result in an effective reduction of *Aedes* populations in several endemic settings [29–31], including Tanzania [32–34]. Within the hotel setting of Zanzibar, we recently observed several *Aedes* species share larval habitats with other vector species as well as nuisance mosquitoes [10]. We argue that the successful control of mosquito populations at hotels in Zanzibar requires the implementation of control measures to target all potential mosquito larval sources. This will include a ban on the use of single-use plastic water bottles due to their status as the most productive mosquito larval habitats when discarded in and around hotel compounds [9, 10]. We encourage the substitution of single use plastic water bottles by onsite water refilling practices and reusable bottle solutions at hotel compounds. In Thailand, this approach has been shown to prevent the use of one million plastic bottles annually, that would otherwise be discarded in the environment [35]. However, the potential impact of this approach on the populations of container-exploiting mosquitoes has, to the best of our knowledge, never been tested before.

Hotels present an ideal environment to test novel mosquito control interventions given the management set up, resources, staff availability and economic incentives for providing a mosquito-safe sustainable environment. However, prior to implementation of large-scale trials, it is crucial to undertake feasibility assessments to determine whether selected intervention approaches are technically, operationally, and socio-economically practical [26, 36, 37]. Feasibility studies form the basis for estimating the required resources (material and manpower) and the cost of a large-scale trial addressing the eco-geographic and environmental spectrum of mosquito occurrence in the hotel setting. Furthermore, they allow for estimation of the level of acceptability and compliance to the LSR approaches by management, staff, and visitors. As such, we propose a tool for evaluating the entomological impact of environmental LSM on the occurrence and abundance of mosquito populations at hotel compounds in Zanzibar through integration of LSR actions and implementation of onsite water-refilling practices. We anticipate that eliminating potential mosquito larval sources from the environment and cutting the production of disposable plastic water bottles will deprive local mosquito fauna of key larval habitats and substantially reduce or eliminate mosquito populations from the hotel compounds.

## Materials and methods

### Study sites

The study hotels have been described in previous surveys [9, 10, 21]. The hotels are in the South-East coastal region of Zanzibar Island, the largest island of Zanzibar archipelago located in the Indian ocean (Fig 1). The island is about 85 km long and 39 km wide, and occupies a total surface area of 1,464 km$^2$ [38]. The topography is mainly dominated by lowland areas, with the highest point at 120 metres above sea level [38]. According to 2022 population census, an estimated 1,889,773 people live on the island, corresponding to 1,291 inhabitants/square kilometre [39]. The eastern part of the island is arid and covered in coral rag (rock made of coral) making it attractive and ideal for fishing villages as well as the tourist and recreation industry, representing the third largest economy of the island [40].

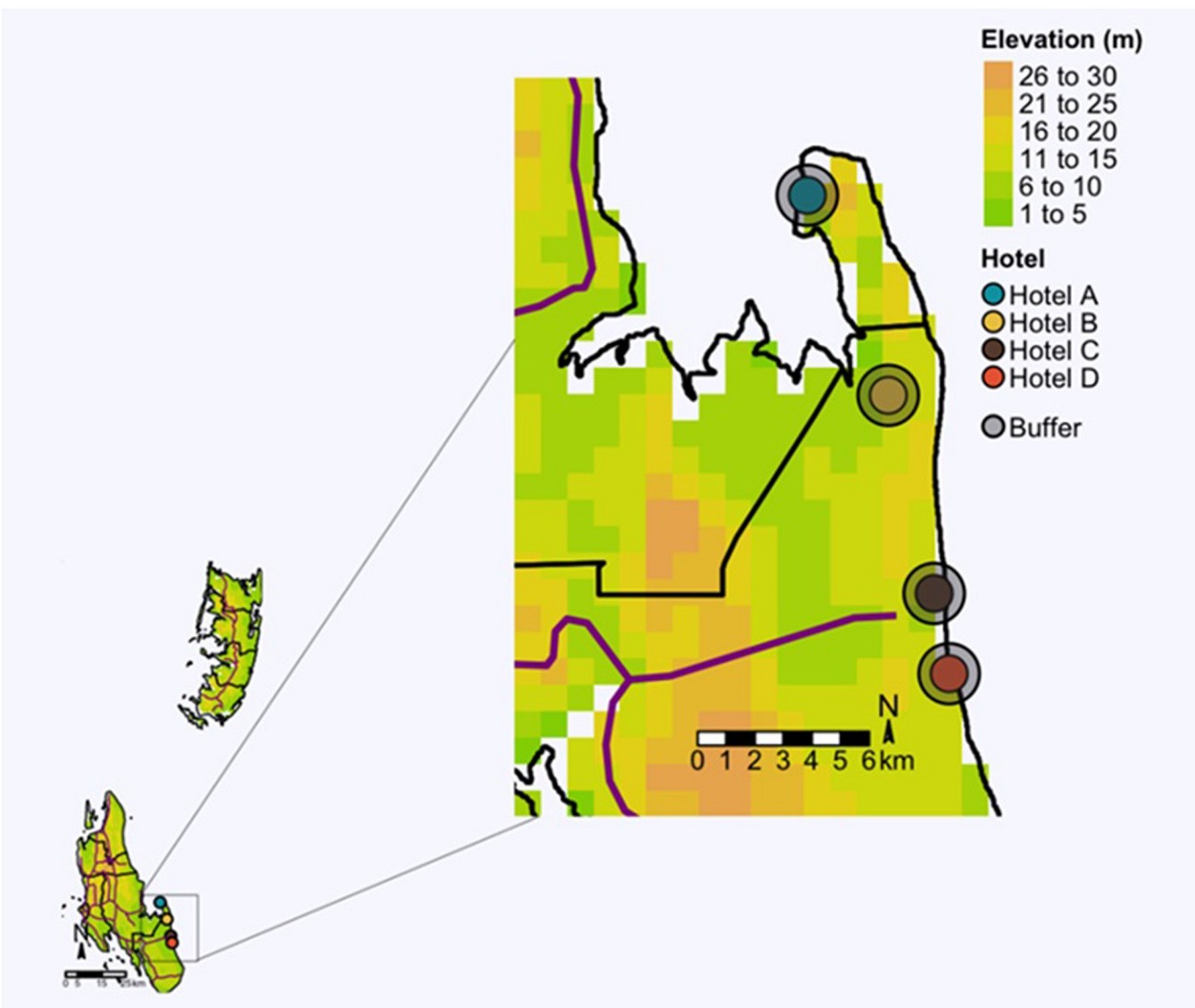

**Fig 1. Location on the east coast of the southern region of Zanzibar (Unguja) Island.** Grey circles around each location indicate a 300 metres buffer radius for deployment of interventions. Dark magenta lines depict main roads. Map made by author. Elevations raster grid and administrative borders shapefile obtained from https://gadm.org, Zanzibar main roads shapefile obtained from http://www.naturalearthdata.com.

*Anopheles arabiensis* is the most dominant malaria vector [41, 42]. However, arboviral disease vectors viz., *Ae. aegypti*, *Ae. bromeliae*, *Ae. metallicus* and *Ae. africanus* also occur. [9, 10, 43]. There are indications of dengue and chikungunya transmission in Zanzibar, but the main vector species are yet to be incriminated. In general, there is no surface rivers in Zanzibar archipelago. The hydrology of the archipelago comprises chains of underground rivers running through as system of caves. The rivers can eventually re-emerge downstream, in some regions, to feed surface pools of water as natural springs [44].

### Recruitment and eligibility

The hotels were selected according to compound size (total residential and non-residential area not less than one hectare), accessibility during lower and higher tourism seasons, willingness to share data and willingness to accept the interventions, and publication of findings.

### Capacity development and training

Actions to reduce mosquito larval habitats need to be integrated into the daily routines of the hotels to ensure sustainability. Therefore, designated hotel staff members will receive a series of tailored, hands-on training prior to and throughout the implementation of LSR control measures. This equips them with the competencies and proper technical skills required to implement control measures. Moreover, it will equip the field staff with skills required to perform monitoring and evaluation of the impact of interventions on larval habitat occurrence and abundance of mosquito immature stages, deployment and maintenance of mosquito traps, and collection of adult mosquitoes, samples preservation, etc. A training syllabus will be co-designed with the Zanzibar Malaria Elimination Programme (ZAMEP) for subsequent use in planned workshops.

### Description of interventions

Based on previous observations at the same hotels [9, 10], the most appropriate mosquito management intervention would be implementation of environmental control measures. Implementation of environmental mosquito measures interventions is supported by the fact that the main mosquito sources found at the hotels were poorly discarded small size artificial and natural containers, such as plastic bottles, coconut shells, automobile tyres, flowerpots, and others [9, 10]. In addition, resistance to DDT and common pyrethroid insecticides is established in the main vector population *Aedes aegypti* [21] suggesting that further use of chemical control may exacerbate insecticide resistance problems. As such, the type of environmental interventions we plan to implement comprise the following two types of actions, that is, 1) weekly removal of discarded artificial and natural containers located in the environment within the hotel compounds and around a 300 metres buffer radius of accessible areas at immediate surrounding communities (Fig 1). The choice of 300 metres radius for the buffer zone is based on the current knowledge of average flight range of Afrotropical mosquito vectors [45–47]. A second type of action to be implemented concurrently with larval habitat removal is deployment of portable water dispensers in most frequented areas within the hotel, such as restaurants, lounges, and entry foyers. Accordingly, reusable glass water bottles will be provided to guests to prevent production of disposable single-use plastic water bottles that may provide suitable habitats for mosquito immatures.

To build trust and reassure guests that the water is safe and encourage them to use the dispenser instead of bottled water, a sign with information regarding water quality and safety will be displayed next to the dispenser. The sign will provide details on where the water comes from, how it is filtered and purified and pH level. In addition, a quick response (QR) code and link that directs guests to the water supplier website will be provided for further details and enquiry on dissolved mineral contents and the standard it meets. Wherever justifiable and logistically practical, larval habitat modification measures, such as covering or plugging of containers (e.g., tree holes), will also be implemented to prevent them from collecting rainfall or any sort of flowing or gardening water that might support mosquito immatures stages.

## Study design

We propose a randomized partial cross-over experimental design to evaluate the impact and feasibility of proposed interventions [48, 49]. The unit of randomization for allocation of interventions is a hotel compound. The intervention is allocated at two of the study hotels while the remaining two will act as controls. The allocation sequence is generated by computer-based randomization. Hotels will be required to stop implementing space spraying activities during the trial period to eliminate a possible confounding factor. A potential replacement for space spraying could be deployment of transfluthrin emanators. Transfluthrin is a fast-acting spatial repellent whose efficacy in reducing the risk of human exposure to malaria, lymphatic filariasis, dengue, and other arbovirus vectors has been demonstrated [50–52]. To ensure compliance in the non-use of plastic water bottles, hotel will be required to implement educative actions. An example of an educative measure to be implemented is the placement of signage inside rooms, and near the dispenser, informing guests and hotel staff about public and environmental health problems associated with plastic use.

## Intervention allocation and impact evaluation

The interventions will be allocated during the dry season (January–March) (Fig 2), when mosquito abundance is predictably lower [9, 10]. The impact on mosquito populations is evaluated throughout the subsequent rainy and dry season for the entire temporal range of mosquito occurrence, as recommended by WHO [26]. Specifically, allocation and evaluation of intervention is conducted in two phases; in the first phase (phase I) it is assigned only to the intervention arm (January–March) and the impact then monitored over six months covering the long rainy season (April–June) and the dry season (July–September). In the second phase (phase II), interventions will be extended to the control arms in October (dry season), provided that an appreciable impact, such as reduction in mosquito density and occurrence probability, is observed at intervention arm during phase I. Then, the impact is to be evaluated again over subsequent rainy and dry seasons (November–May). This will demonstrate whether any potential reduction of mosquito density and occurrence probability observed in the intervention arm can be reproduced and maintained in the control arm.

## Sample size calculation

Provisional sample size is calculated assuming that experiments are expected to be carried for 24 months. At each control and intervention hotel, mosquito populations data collections will be undertaken for 7 days per month. Therefore, we expect to have at least n = 168 replicate-days of observation per hotel, which would correspond to a provisional total minimum sample size of N = 168 days x 4 hotels = 672 days of observations for the entire study.

## Statistical power estimation

Considerable complexity is associated with this type of field trial, including the existence of fixed (e.g., intervention) and random effects (site and number of observations and replicates) and possible hierarchical dependence between observations across sites over time. To account for all factors, the final sample size and statistical power needed for the study were estimated using simulations, following the approach proposed by Bolker [53]. The statistical power was estimated considering that an initial minimum number of 168 replicate-days/site would be enough to detect a minimum reduction of 40% (effect size) in relative risk of mosquito abundance at treated hotels (intervention arm) compared to untreated (control arm) hotels. The effect size was set based on findings from a larger and multicentric larval source management

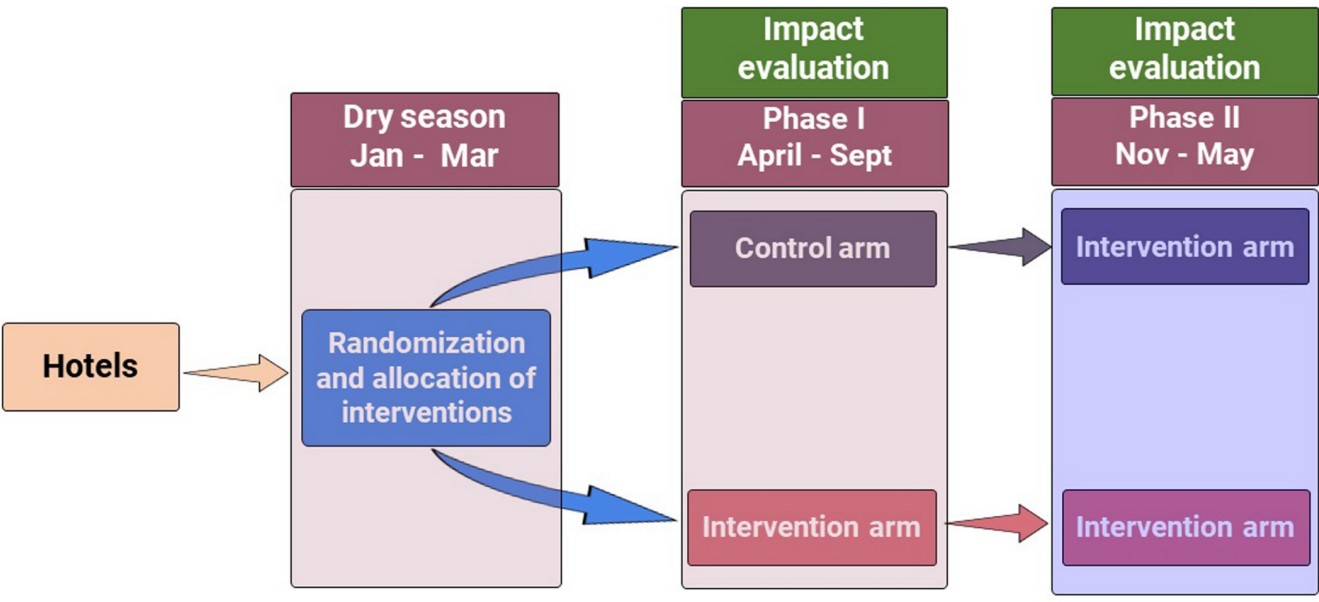

**Fig 2. Schematic representation a randomised partial cross-over experimental design for evaluating the impact of larval source management interventions at hotel setting in Zanzibar.**

randomized trial conducted in Nicaragua and Mexico [54]. Accordingly, the overall mean variation of mosquito counts between seasons, daily replicates and hotels were estimated according to a previous survey at the same hotels as being, respectively, 18.5, 2, and 5 [10]. The daily mosquito counts were assumed to follow a negative binomial distribution with overdispersion in mosquito counts of k = 0.5. This information was applied to simulate the expected spatio-temporal distribution of mosquito counts according to type of intervention. For this particular purpose, Hotel A and Hotel B were arbitrarily considered as being treated, whereas Hotel C and Hotel D functioned as control. The generated dataset with factors, namely hotel, day, month, season and intervention type, was then used to fit a probable effect of interventions on simulated mosquito counts (i.e., response variable) by using generalized linear mixed models (GLMM) at α = 5% significance level. The resulting model was iteratively fitted 1000 times by means of Monte Carlo simulations via the R software package simr v. 1.0.5 [55]. Here we gradually increased the number of observations within hotels and extracted the p–values from fitted models (see Fig 3). Models were fitted to address the hypothesis that there would be no significant reduction in overall mosquito counts in the intervention arm compared to the control. The mean proportion of the time this null hypothesis was rejected is then the statistical power [53]. Simulation results suggest that at least 142 sampling days are required per site for the entire period planned for the experiment. This would confer a satisfactory statistical power of 81% (CI = 79%– 83%) (Fig 3). A script for simulating statistical power using R software is provided as S1 File.

## Data collection procedures

### Characterization of mosquito larval sources

In each hotel, mosquito surveys will be undertaken for seven days per month over 142 days. Outdoor spaces of the hotel compounds will be inspected for the presence of immature mosquito stages. Only containers holding water will be examined to record the presence of immature

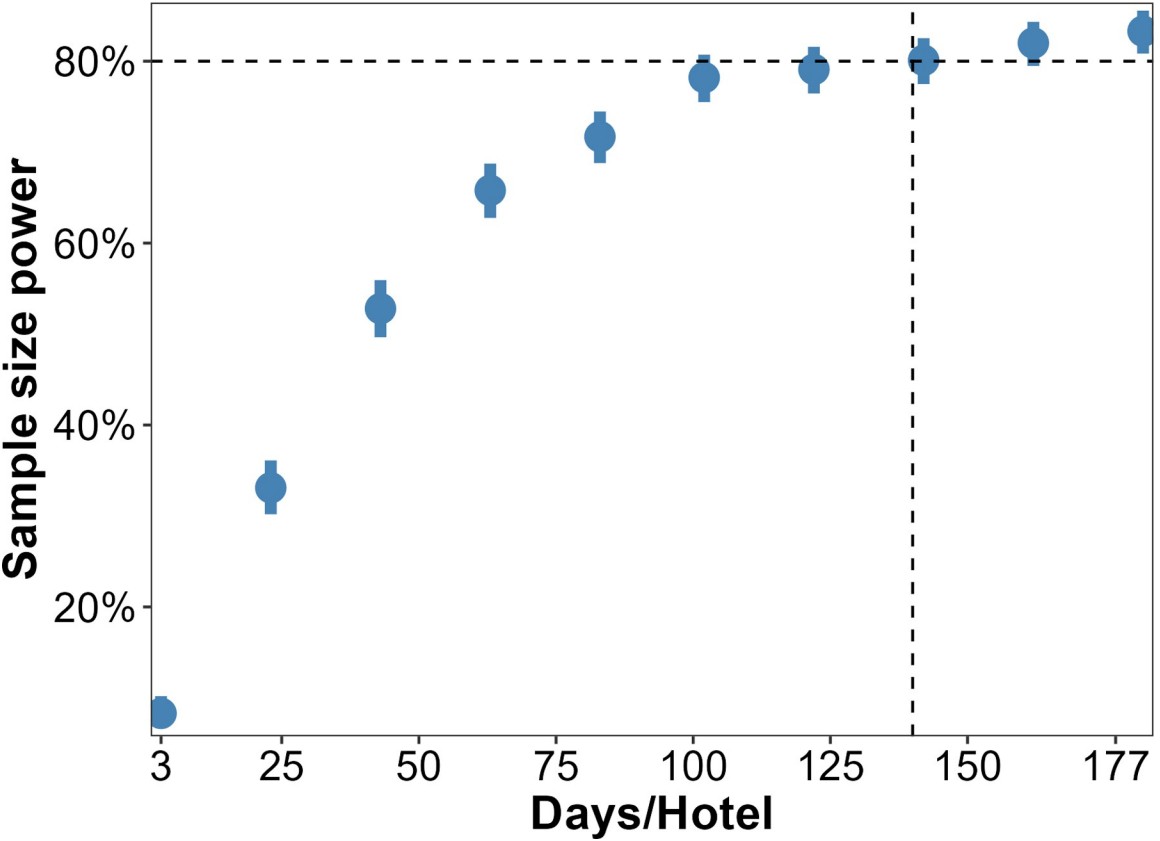

**Fig 3. Inferred sample size required to detect significant variations of mosquito abundance associated with introduction of larval sources management interventions at hotels.** A minimum 142 replicate-days of observations are required at each of the four experimental sites.

mosquitoes. Mosquito larval sources will be divided according to type (i.e., natural and artificial containers) and sub-classified according to physical characteristics, function, location and dimension (Table 1), as suggested by Flaibani et al., [56] and used in our previous findings [9].

### Immature mosquito sampling strategy

The following sampling strategy adapted from Knox et al., [57], Manrique-Saide et al., [58] and Tun-Lin et al., [59] will be applied for the collection of mosquito larval data within the hotel compounds and immediate surrounding areas. For containers of less than 20 litres of water (e.g., bottles, snail shells, flasks, tins, tyres, tree holes, and coconut husks, etc.), all the larvae and pupae are collected by sieving the water directly through a mesh net or by a pipette. If water volume is more than 20 litres and emptying the container is not feasible (due to its size or nature of the container) but there is good visibility (tank/drums, pots, swimming pools, tubs, etc.) and low larval and pupal density (less than 100 individuals), then all the specimens are sampled using dippers and sweep nets. Otherwise, only 10–15 dips/sweeps are taken at random to sample immature stages and then a correction factor is applied to estimate density. If containers are too deep (e.g., underground wells), samples are collected using small buckets (approx. 5 litres) suspended by a rope. Artificial containers found with water, but no larvae (on the day of sampling) are considered as potential mosquito larval sites if it they have physical conditions to retain collected water for at least three consecutive days. Samples are marked

**Table 1. Proposed scheme for characterization of containers supporting the development of mosquito larvae according to physical and functional features, and location in the landscape.** Adapted from Flaibani et al., [56] and Kampango et al., [9].

| Characteristic | Categories | Description |
| --- | --- | --- |
| Physical | Type | Cans/Tins |
| | | Bottles |
| | | Buckets |
| | | Dishes |
| | | Flowerpots |
| | | Jars |
| | | Fast food containers |
| | | Swimming pools |
| | | Tanks |
| | | Wells |
| | | Coconut shells/Flower spathes |
| | | Mollusc shells |
| | | Tree holes |
| | | Fallen leaves |
| | | Tyres |
| | Material | Plastic |
| | | Clay |
| | | Metal |
| | | Rubber |
| | | Concrete |
| | | Glass |
| | Size/Volume | Small ($< 1$ litres) |
| | | Medium ($>1$ litres and $<5$ litres) |
| | | Large ($> 5$ litres and $<10$ litres |
| | | Very larger $> 10$ litres |
| Function | Purpose | Gardening |
| | | Water storage |
| | | Animal feeding |
| | | Recreation |
| | Use | In use |
| | | Not in use |
| Location | Environment | Staff room quarter |
| | | Guest room quarter |
| | | Administrative area |
| | | Restaurant/Bar area |
| | | Dumpsite |
| | | Kitchen/Laundry area |
| | | Garden/Open spaces |
| | | Plant nursery |
| | Sunlight exposure | Not exposed |
| | | Exposed |

according to collection day and location and transported to an insectary for further processing. Larvae of mosquito species larvae that are known to be predacious, such as, *Toxorhynchites* species, *Cx. tigripes* and *Eritmapodites* species, need to be separated into different vials to avoid

predation on other mosquito larvae. An extra sample of prey mosquito larvae is collected to feed the predacious ones.

## Adult mosquito sampling strategy

A survey of adult mosquito populations will be undertaken concurrently with larval surveys to assess whether any impact on immature mosquito stages is translated into lower adult mosquito populations and, consequently, reduction in bite exposure. Therefore, outdoor host-seeking mosquitoes will be sampled for 24-hours using BG Sentinel traps (Biogent, Germany) baited with $CO_2$ and CO2 –baited light traps. Ligh trap collections will be performed from 17:00–06:00 hours. Additionally, outdoor resting mosquitoes will also be collected using Prokopack aspirators (John W. Hock Company, USA) between 06:00–09:00 hours.

## Sample processing and identification

Following the procedure described in [10], collected samples of immature mosquitoes will be reared under insectary environmental conditions of temperature (27 ± 2˚C) and relative humidity (75 ± 10%) until they emerge as adults [60]. Larvae will be fed either with ground adult cat food biscuits (non-*Anopheles* mosquito larvae) or with Koi fish food (*Anopheles* larvae) and kept at a 12:12h light: dark regime until adult emergence. Adult mosquitoes will be euthanised by placing the specimens inside a refrigerator. Dead mosquitoes will be grouped in batches of ten to twenty specimens, and then transferred into Eppendorf tubes containing silica gel for morphological identification. Morphological identifications will be done using taxonomic keys for Afrotropical mosquito fauna [61–65]. Mosquitoes that cannot be identified morphologically to species, such as members of species complexes, will be placed individually in Eppendorf tubes with silica gel and store at -20˚C for further molecular analysis.

## Monitoring and evaluation of intervention impacts

Potential impacts of the interventions will be observed monthly during the trial period. The monitoring and evaluation process aim to measure the trend of the following entomological indicators in both control and intervention arms, namely: mosquito immature occurrence and abundance, and type, frequency, and productivity of larval habitats. For *Aedes* mosquitoes, some *Aedes* (*Stegomyia*) immature indexes, namely container index and pupae will also be assessed. Since consistent reduction of mosquito larval stages is not itself a sufficient indicator of the success of control interventions [26], the impact of the interventions on adult mosquitoes occurrence and abundance also needs to be measured.

## Evaluation of the use and of water dispensers

A questionnaire survey to evaluate the level of acceptability and use of water dispensers will be carried out. Therefore, guests will be asked to answer to a questionnaire to determine the level of usage, preference, and satisfaction in using the water dispenser. The questionnaire is also intended to obtain suggestion for improvement. The questionnaire will ask question such as, how often do you use the water dispenser? At what times of the day do you typically use the water dispenser? Why do you choose to use the water dispenser? If you do not use the water dispenser, why not? Are you satisfied with the quality of the water from the dispenser? Are you satisfied with the location of the water dispensers in the hotel? What improvements would you suggest for the water dispenser service? Demographical information namely age, gender and provenance of guests will also be recorded.

## Quality assurance assessment of control operations and staff performance

Periodic evaluation of the quality of application and execution of mosquito control intervention by trained field staff by a senior supervisor or an independent group of accessors is needed. This ensures that interventions have been applied correctly and at the appropriate time, whether control operation staff work in designated areas, that habitats have been searched for immatures and adult mosquitoes, and that the interactions between control staff, hotel staff and hotel guests support the goals of the project. This information will be shared with project coordinators for analysis and implementation of any corrective measures that are required.

## Data analysis

Joint species distribution models (JSDMs) are appropriate for evaluating the response of mosquito communities to intervention [66]. The JSDMs will be fitted using the Bayesian hierarchical model of species community (HMSC) approach proposed by Ovaskeine et al., [67]. Changes in both mosquito community occurrence and abundance in response to intervention can be assessed with these models. JSDMs will be fitted using the software Hmsc v. 3.0–12 [68]. For modelling the mosquito community occurrence, mosquito counts will be converted into a presence (Y = 1) and absence (Y = 0) binary matrix. The presence-absence observation will be assumed to follow a probit distribution. For the abundance model, abundance conditional on presence modelling approach will be applied. To do so, count data is transformed by declaring zeros as missing data, log-transformed, and then scaled to zero mean and unit variance for each species. The log-transformed abundance will be assumed to follow a Gaussian distribution. Modelling abundance requires more information. Therefore, only a subset of species that were observed in at least ten sampling occasions will be used to model the impact of the intervention on abundance. A similar approach has been applied elsewhere [69]. Together with the presence or absence of interventions, season will also be added to the models as a fixed effect for its significant effect on mosquito occurrence and abundance as described in previous surveys at same sites [9, 10]. Site and day of survey will be included as random factors to account for unmeasured variability in mosquito occurrence and abundance across site over time. The duration of time (in days) between sampling occasions in a given site will be considered as a latent temporal random covariate to account for temporal dependence between repeated measure, as suggested by Ovaskaine and Abrego [69]. We are assuming that spatial dependence between residuals of collected data is negligible as study hotels are located at a relatively large distance, at least 1.8 km, between one to another. Additionally, the models will be phylogenetically corrected to explore whether closely related species have similar responses to the interventions than distantly related species. The phylogenetic tree used in the analyses will encompass six levels, ranging from mosquito species order, family, subfamily, genus, subgenus, and species and be derived from data published by [70]. The R software package ape v. 5.5 will be used to convert the taxonomic tree into a matrix, assuming equal branch lengths among families, among genera within a family, and among species within a genus [69]. All the statistical analysis will be performed using the latest available version of the open source domain software R [71].

## Data management

Standard field data collection methods will align with those used in previously published observational studies [10]. The collected data will include mosquito larval habitat type, applied interventions, presence and absence of mosquito larvae, and number of visited sites. Approved data will be entered into password secured Excel spreadsheets for future analysis and

preparation of feedback reports to project coordinators and partners. This study is not a clinical trial evaluating a therapy with mortality or irreversible morbidity endpoints, so no data safety monitoring board is needed.

## Primary outcomes

The primary outcomes of the project will be the efficacy of interventions on mosquito occurrence, abundance, and composition of mosquito assemblages in the intervention arm compared to the control arm.

## Secondary outcomes

Secondary outcomes of the project will be the characterization of larval habitats, according to type, function, and location as detailed in Table 1.

## Advocacy and communication

Prior to implementation of the study, a communication plan will be produced together with the hotel management, in collaboration with other partner institutions, to attract support from guests, the local community and relevant actors/stakeholders including the Zanzibar Association of Tourism Investors (ZATI), environmental authorities, and Zanzibar Ministry of Health through the ZAMEP. ZAMEP and other identified institutions, as well as non-Governmental Organizations (NGOs), involved in policy making, decision-making and implementation of MBD control interventions in the archipelago will be approached to identify options for collaboration and sharing of resources to improve efficiency and optimize benefits for public health.

## Environmental impact assessment

Environmental management interventions are expected to reduce the impact on the environment of vector control [36]. However, some operations may affect certain non-target organisms. Therefore, prior to implementation of interventions, consultation with experts from Zanzibar Environment Management Authority (ZEMA) will be needed to identify any potential protected plants or animals occurring within the studied areas and get advice on how to safeguard the integrity of the organism in question. Alternatively, if no specialist is found, an environmental impact assessment matrix [36] will help to identify and minimise potential non-target effects.

## Ethical considerations

Ethical clearance for the proposed trial has been granted by the Zanzibar Health Research Ethics Review Committee (ZAHREC), Ref: No. ZAHREC/03/PR/Oct/2019/001, in the Second Vice President's Office, as part of the project Environmental Sustainability of Hotels on Zanzibar (EnSUZA). Written informed consent to conduct the experiment at hotel compounds was obtained from each hotel manager. Additional permissions to carry out the study were obtained from local community leaders (*Shehia*) responsible for the area where the study will be conducted.

## Study status

Field experiments to assess the impact of proposed interventions had been planned to be carried out in 2021. However, the protocol did not move on to field testing due to global restrictions

imposed by the COVID-19 pandemic which had dramatically affected the tourism industry in Zanzibar, forcing some hotel partners to close doors from mid-2020 throughout 2021.

## Data availability

The underlying data will be shared via the Open Science Framework (OSF) repository. Data will be available under the terms of the Creative Commons Zero "No rights reserved" data waiver (CC0 1.0 Public domain dedication).

## Discussion

With the advent of DDT, global efforts to control MBDs by mean of environmental management approaches shifted radically to the widespread use of synthetic chemical insecticides [72]. However, emergence and widespread resistance in vector species worldwide, coupled with resurgence of MBDs in regions where it had earlier been eradicated is a clear reminder that effective elimination of MBDs may not occur by relying only on stand-alone insecticide-based interventions. In fact, historical [29, 30, 73–76] and contemporary evidence [33, 77, 78] show that effective MBD elimination programmes were achieved by integrated implementation of vector control approaches that relied fundamentally on mosquito larval source management interventions.

The main factors that have discouraged wide-scale use of environmental management interventions (EMI) for vector control are the initial implementation costs and the challenges related to scaling-up interventions to cover large areas in a short timeframe [79]. However, systematic analysis has shown that EMI can be economically cost-effective in the long-term due to the longevity of its protection [75, 80]. Moreover, it can reduce the likelihood of selection for insecticide resistance [36, 75, 81]. Perhaps two of the most notorious cost-effectiveness analyses of EMI conducted in the sub-Saharan region come from studies on malaria control in copper mines in Zambia [82] and at a military base in Apapa, Nigeria [83]. In Zambia, the full package of control measures consisted of vegetation clearance, modification of riverbanks, draining swamps, oil application to open water bodies and house screening [82], whereas in Nigeria it consisted of swamp drainage [83]. Malaria incidence was reduced by 70% to 95% in Zambia and Nigeria, respectively, after the introduction of these control measures. Moreover, the interventions averted 4,173 deaths and 161,205 malaria attacks in the Zambia study. EMI can thus control a wide range of vectors and nuisance pests sharing the same breeding sources. Evidence with malaria vectors control interventions have shown that although initial cost of environmental management may be high, particularly during the installation of infrastructure, long term maintenance costs can be exceptionally low while the efficacy of control measures can prevail for decades [75].

The protocol that we propose will assess the impact of integrating larval source reduction and installation of water dispensers on vectors and nuisance mosquito species at hotel compounds in Zanzibar. We anticipate that combining these types of controls in one intervention is more likely to cause meaningful reduction of vector species than if each was evaluated separately. This innovative approach is less environmentally destructive than other EMI interventions and may help to break the loop of production of single-use plastic water bottles that are discarded and eventually become mosquito larval habitats at hotel compounds [9, 10]. However, the success and sustainability of this type of hotel-based mosquito management approach depends on the compliance of studied hotels. Therefore, to minimize the risk of non-compliance, and its potential impact on mosquito occurrence and abundance, we suggest that the intervention is performed in daily coordination with designated hotel staff members. Another challenge this type of study would face in some locations is the establishment of buffer zones,

since some hotels are immediately surrounded by other hotel compounds, human settlements, or by nearly impenetrable forested areas. Therefore, extending intervention to such areas may not be feasible.

Our protocol involves both contemporary and historical controls. This is expected to reduce the between-hotel variability, permitting the detection of smaller effect-sizes (in this case reduction in mosquito probability of occurrence and abundance) with relatively low investment in resources [48, 84]. Moreover, partial cross-over design can have similar efficiency as a complete cross-over design in answering complex questions asked in a feasibility trials proposed here [48]. Results from this trial will support a decision-making process exercise and determine whether to proceed to large scale phase.

Worldwide, tourism destinations have been confronted by an unprecedented plastic crisis due to proliferation of single use plastic water bottles [85]. This has led to calls for global sustainable solutions that reduce or totally replace disposable plastic bottles. This has also been acknowledged in the Zanzibar Blue Economy Policy [86]. Perhaps one of the most cost-effective sustainable approaches that has received attention in the tourism industry are onsite water refilling practices [87, 88]. In Thailand, implementation of onsite water refilling policies in a complex of hotels, resorts and spas has prevented the disposal of 1.8 million single-use plastic water bottles per year [35]. The impact of type of policy change on container-breeding mosquito population has yet to be tested addressed. This protocol can form the base for future studies aimed to evaluate the impact of environmental control interventions targeting the risk of mosquito exposure at hotel compounds.

## Supporting information

**S1 File. R software code script used to simulate appropriate sample size and statistical power of the sample size.**
(R)

## Acknowledgments

We thank hotel partners for accepting to be part of the study. Thanks are due to the Zanzibar Association of Tourism Investors for embracing the project and for the encouragement to perform this study. We would like to extent our appreciation to the reviewers for their insightful comments and suggestions.

## Author Contributions

**Conceptualization:** Ayubo Kampango.

**Formal analysis:** Ayubo Kampango.

**Funding acquisition:** Peter Furu.

**Methodology:** Ayubo Kampango, Fatma Saleh, Peter Furu, Flemming Konradsen, Michael Alifrangis, Karin L. Schiøler, Christopher W. Weldon.

**Project administration:** Peter Furu.

**Resources:** Fatma Saleh, Peter Furu, Flemming Konradsen, Michael Alifrangis, Karin L. Schiøler, Christopher W. Weldon.

**Software:** Ayubo Kampango.

**Supervision:** Peter Furu, Michael Alifrangis, Karin L. Schiøler, Christopher W. Weldon.

**Validation:** Ayubo Kampango, Fatma Saleh, Peter Furu, Flemming Konradsen, Michael Alifrangis, Karin L. Schiøler, Christopher W. Weldon.

**Visualization:** Ayubo Kampango.

**Writing – original draft:** Ayubo Kampango.

**Writing – review & editing:** Ayubo Kampango, Fatma Saleh, Peter Furu, Flemming Konradsen, Michael Alifrangis, Karin L. Schiøler, Christopher W. Weldon.

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
