## [Decision Letter · Decision Letter 0]

29 Aug 2023

PONE-D-23-14190A protocol for evaluating the entomological impact of larval source reduction on mosquito vectors at hotel compounds in ZanzibarPLOS ONE

Dear Dr. Kampango,

Thank you for submitting your manuscript to PLOS ONE. After careful consideration, we feel that it has merit but does not fully meet PLOS ONE’s publication criteria as it currently stands. Therefore, we invite you to submit a revised version of the manuscript that addresses the points raised during the review process.

We look forward to receiving your revised manuscript.

Kind regards,

David Zadock Munisi, Ph.D

Academic Editor

PLOS ONE

Journal Requirements:

“Preparation of this study protocol is supported by Denmark’s Development Corporation (Danida), through the Danida Fellowship Centre (DFC), as part of the EnSuZa project Grant 17-04-KU. Additional support was obtained from the Danida’s Building Stronger Universities Phase III programme at the State University of Zanzibar (SUZA).”

“Preparation of this material is supported by Denmark’s Development Corporation (Danida), through the Danida Fellowship Centre (DFC), as part of the EnSuZa project Grant 17-04-KU. Additional support was obtained from the Danida’s Building Stronger Universities Phase III programme at the State University of Zanzibar (SUZA).”

“Preparation of this study protocol is supported by Denmark’s Development Corporation (Danida), through the Danida Fellowship Centre (DFC), as part of the EnSuZa project Grant 17-04-KU. Additional support was obtained from the Danida’s Building Stronger Universities Phase III programme at the State University of Zanzibar (SUZA).”

Reviewers' comments:

Reviewer's Responses to Questions

**Comments to the Author**

1. Does the manuscript provide a valid rationale for the proposed study, with clearly identified and justified research questions?

Reviewer #1: Yes

Reviewer #2: Yes

Reviewer #3: Partly

Reviewer #4: Yes

2. Is the protocol technically sound and planned in a manner that will lead to a meaningful outcome and allow testing the stated hypotheses?

Reviewer #1: Partly

Reviewer #2: Partly

Reviewer #3: Partly

Reviewer #4: Yes

3. Is the methodology feasible and described in sufficient detail to allow the work to be replicable?

Reviewer #1: Yes

Reviewer #2: No

Reviewer #3: No

Reviewer #4: Yes

4. Have the authors described where all data underlying the findings will be made available when the study is complete?

Reviewer #1: Yes

Reviewer #2: No

Reviewer #3: No

Reviewer #4: No

5. Is the manuscript presented in an intelligible fashion and written in standard English?

Reviewer #1: Yes

Reviewer #2: Yes

Reviewer #3: Yes

Reviewer #4: Yes

6. Review Comments to the Author

You may also provide optional suggestions and comments to authors that they might find helpful in planning their study.

Reviewer #1: Review of “A protocol for evaluating the entomological impact of larval source reduction on mosquito vectors at hotel compounds in Zanzibar”

The manuscript is well-written, clear and concise in its presentation of a study protocol for mosquito-borne disease control in a tourist setting. Editing of occasional spelling mistakes and minor grammatical errors would help improve presentation in some places. The writing is very good overall. Comments on the protocol focus on the statistical aspects.

Comments:

p. 7: “The choice of 300 metres radius for the buffer zone is based on the current knowledge of maximum flight range of Afrotropical Aedes mosquito vectors …” -- One of the studies cited in support of this statement is Marcantonio et al. (2019), who report maximum observed dispersal distance for Aedes aegypti greater than 680 metres. Do the authors intend to refer to mean dispersal estimates rather than maximum?

P. 8-9: The section “Statistical power estimation” investigates the statistical power to detect an effect. It appears that the power estimation only applies to the first phase of the intervention (see section “Intervention allocation and impact evaluation”). The analysis appropriately accounts for the hierarchical data structure and non-normal response using generalised linear mixed models. However, the analysis only reports the minimum number of replicate-days required to achieve sufficient power. It would be of interest for the power analysis to also compare the number of hotels involved in the trial, given the relatively low number of hotels in the design (two treated, two control in power analysis). Also, the protocol discusses the advantage of partial cross-over studies but this aspect does not appear to be represented in the power analysis.

p. 11: The “Data analysis” section presents a brief sketch of the software packages proposed for analysis of future data. The software package Hmsc can be used to fit a variety of models, and it would help to add clarification of the actual statistical model that will be used. For example, count data is proposed to be modelled conditional on observed presences after a log transformation. It is presumed that the likelihood will be Gaussian, but this is not stated. The authors could also clarify if the analysis will account for spatial dependence by a Gaussian process in the proposed joint species distribution models.

Reviewer #2: Comments

Find below my comment

The introduction seems to me too long may be it should be shortened a bit

Sample size calculation

What should be considered as a unit for comparison is it the hotel or something else I didn’t got this very well in the paper. I think the author should explain 4 hotels seems to me too small.

The presence of water attraction facilities such as swimming pools (number and size)… Should be recorded since these could serve as breeding environment for mosquitoes. The average number of people living in each hotel should also be considered.

Study design

The authors did not indicated whether they are going to collect baseline data and for how long this is very important because it will enable to avoid bias in allocating cluster and this data will be important to assess the impact of the intervention and monitor temporal or seasonal variations.

The author indicated the following “we plan to implement comprise the following two types of actions, that is, 1) weekly removal of discarded artificial and natural containers located in the environment within the hotel compounds and around a 300 metres buffer radius of accessible areas at immediate surrounding communities (Figure 1). The choice of 300 metres radius for the buffer zone is based on the current knowledge of maximum flight range of Afrotropical Aedes mosquito vectors [46, 47, 48]. A second type of action to be implemented concurrently with larval habitat removal is deployment of portable water dispensers in most frequented areas within the hotel, such as restaurants, lounges, and entry foyers. Accordingly, reusable glass water bottles will be provided to guests to prevent production of disposable single-use plastic water bottles that may provide suitable habitats for mosquito immatures.” I am afraid that they fact that the authors are not considering also controlling mosquito in standing water collections which are ideal habitats for anopheline and some culicine could affect the efficacy of the intervention”

The authors should consider adding a method of evaluation of the use of water dispensers by people living in the hotel to determine if the level of usage of these measures is similar between hotels because this could introduce some bias in the analysis.

The author indicated using BG sentinel traps for collecting adult mosquito I am afraid this method alone will be less efficient to collect adult mosquitoes I will advise them to use two or three different collection methods to increase chances of evaluating mosquito biting behaviour.

Reviewer #3: Unfortunately just proposed work in the form of a protocol does not constitute as a manuscript in my opinion.

The introduction and the protocol is well written but without any solid data it does not qualify as a research paper.

Reviewer #4: Larval source reduction is very important mosquito control strategy particularly with the emergence of insecticide resistance among vector mosquitoes. Thus, making this protocol very important. However, detailed information on the study site is needed to understand the practicability of the protocol.

Are there any settlements around the study hotels?

Will there be an intervention in a settlement found within 300 m radius buffer zone.

…“Since modelling abundance requires more information, it is proposed that a subset of species that were observed in at least ten sampling occasions be used to model the impact of the intervention on abundance” …..

Will Aedes species be the only target group to measure the impact of the intervention? If not, Anopheles and other mosquitoes have about 1-2 km flight range, how would mosquitoes that will come from breeding areas outside the hotel and the buffer zone be accounted for?

Sanitation as a tool to control Aedes mosquitoes is not new. Compliance is always the problem. How will you ensure compliance among the hotel staffs and guest. For example, how would ensure that guest drink from the water dispenser but not bottled water?

The authors rightly indicated some of these questions as limitations but did not give detailed mitigation measures. However, without proper mitigation measures this protocol might not give the desirable results.

7. PLOS authors have the option to publish the peer review history of their article (what does this mean?). If published, this will include your full peer review and any attached files.

Reviewer #1: No

Reviewer #2: No

Reviewer #3: No

Reviewer #4: No

---

## [Author Response · Author response to Decision Letter 0]

22 Sep 2023

Thank you for you comments and suggestion. 

Please find below the response to your concerns.

Reviewer #1: Review of “A protocol for evaluating the entomological impact of larval source reduction on mosquito vectors at hotel compounds in Zanzibar”

The manuscript is well-written, clear, and concise in its presentation of a study protocol for mosquito-borne disease control in a tourist setting. Editing of occasional spelling mistakes and minor grammatical errors would help improve presentation in some places. The writing is very good overall. Comments on the protocol focus on the statistical aspects.

Comments:

p. 7: “The choice of 300 metres radius for the buffer zone is based on the current knowledge of maximum flight range of Afrotropical Aedes mosquito vectors …” -- One of the studies cited in support of this statement is Marcantonio et al. (2019), who report maximum observed dispersal distance for Aedes aegypti greater than 680 metres. Do the authors intend to refer to mean dispersal estimates rather than maximum?

R: Thanks for the comments. We are referring to average dispersal range. This information has been amended.

P. 8-9: The section “Statistical power estimation” investigates the statistical power to detect an effect. It appears that the power estimation only applies to the first phase of the intervention (see section “Intervention allocation and impact evaluation”). The analysis appropriately accounts for the hierarchical data structure and non-normal response using generalised linear mixed models. However, the analysis only reports the minimum number of replicate-days required to achieve sufficient power. It would be of interest for the power analysis to also compare the number of hotels involved in the trial, given the relatively low number of hotels in the design (two treated, two control in power analysis). Also, the protocol discusses the advantage of partial cross-over studies, but this aspect does not appear to be represented in the power analysis.

R: Thanks for your concern. In summary, statistical power was estimated for the entire period planned for undertaking the study, that is, 24 months. Hotels were also considered in estimation of statistical power. Hotel variable was considered as factor, together with season, and intervention type. This information was used to generate a dataset containing expected distribution of mosquito community abundance, assuming negative binomial distribution of mosquito counts and overdispersion of k = 0.5. While generate the dataset, we have conservatively assumed that the intervention may cause reduction of 40% in overall mosquito abundance at intervention arm compared to control arm. Generated dataset was then used to fit the probable effect of interventions on simulated mosquito counts (i.e., response variable) by using generalized linear mixed models (GLMM). The resulting model was iteratively fitted 1000 times by means of Monte Carlo simulations. Then, 1000randomly generated models were fitted to address the hypothesis that there would be no significant reduction in overall mosquito counts in the intervention arm compared to the control. The mean proportion of the time this null hypothesis was rejected is then the statistical power. Please refer to methods subsection describing statistical power simulation for further details.

p. 11: The “Data analysis” section presents a brief sketch of the software packages proposed for analysis of future data. The software package Hmsc can be used to fit a variety of models, and it would help to add clarification of the actual statistical model that will be used. For example, count data is proposed to be modelled conditional on observed presences after a log transformation. It is presumed that the likelihood will be Gaussian, but this is not stated. The authors could also clarify if the analysis will account for spatial dependence by a Gaussian process in the proposed joint species distribution models.

R: The paragraph has been amended to include the following statement. “log transformed mosquito counts are assumed to follow Gaussian distribution. 

Thank you also for asking about how we are going to deal with spatial structure between data from different sites. Actually, we thought about that too. However, as it can be seen by the map of study sites submitted, the minimum distance between one hotel and another is least 1.8 km. Therefore, we believe that the risk of spatial dependence caused by, for instance, migration of mosquito populations among the four hotels may be negligeable. Regarding applying gaussian process model to account for spatial dependence, after a thorough consultation of specialized literature, we think for our case, that is, four hotels located at relatively large distance, applying gaussian processes models, such as (GMRF) would not accurately account for spatial dependence. According to Matsuura (2023), the number of locations should be large enough to capture the spatial variability of the data. However, there is no consensus on the minimum number of locations suitable to apply spatial structured models. 

Matsuura, K (2023). Spatial Data Analysis Using Gaussian Markov Random Fields and Gaussian Processes. In Bayesian Statistical Modeling with Stan, R, and Python. Springer. pp 285–329

Differently from spatial dependence, temporal dependence between repeated measures made at a same hotel over time may certainly occur. To deal with possible temporal structure between mosquito counts residuals, the duration of time (in days) between sampling occasions will be included in the models as a temporal random covariate.

Reviewer #2: Comments

Find below my comment

The introduction seems to me too long maybe it should be shortened a bit.

R: Thanks, we have tried to shorten the introduction a little bit more.

Sample size calculation

What should be considered as a unit for comparison is it the hotel or something else. I didn’t got this very well in the paper. I think the author should explain 4 hotels seems to me too small. 

R: As stated in the method section, entomological indicators will be compared between control and intervention arms. We have acknowledged in discussion that the small number of hotels is a constraint as it prevents us from make wide generalization beyond the studied hotels. 

The presence of water attraction facilities such as swimming pools (number and size). Should be recorded since these could serve as breeding environment for mosquitoes. The average number of people living in each hotel should also be considered. 

R: As far as we are concerned there is no artificial public water attraction, notably swimming pools, in the study region. Swimming pools are mostly private and usually found inside private properties. As we have stated in methods all type of containers capable of collecting water for more than 3 consecutive days will be inspected for mosquito immature stages and recorded. 

Study design

The authors did not indicated whether they are going to collect baseline data and for how long this is very important because it will enable to avoid bias in allocating cluster and this data will be important to assess the impact of the intervention and monitor temporal or seasonal variations.

R: Baseline data were collected in previous studies carried out at the same hotels. The findings from baseline studies have already been published (https://doi.org/10.1111/mve.12525, https://doi.org/10.1186/s13071-021-05005-9, https://www.ncbi.nlm.nih.gov/pubmed/35576233). The information has helped in the design of this study protocol. 

We talked about baseline data in Introduction. “In previous studies [9, 10, 21], we have shown that hotels in Zanzibar may provide suitable conditions for sustaining mosquito vectors of global concern, notably Ae. aegypti, Ae. bromeliae, Ae. africanus, Ae. metallicus, Cx. quinquefasciatus, Er. quinqueviattatus, Er. chrysogaster and, to less extent, An. gambiae [9, 10]”

The author indicated the following “we plan to implement comprise the following two types of actions, that is, 1) weekly removal of discarded artificial and natural containers located in the environment within the hotel compounds and around a 300 metres buffer radius of accessible areas at immediate surrounding communities (Figure 1). The choice of 300 metres radius for the buffer zone is based on the current knowledge of maximum flight range of Afrotropical Aedes mosquito vectors [46, 47, 48]. A second type of action to be implemented concurrently with larval habitat removal is deployment of portable water dispensers in most frequented areas within the hotel, such as restaurants, lounges, and entry foyers. Accordingly, reusable glass water bottles will be provided to guests to prevent production of disposable single-use plastic water bottles that may provide suitable habitats for mosquito immatures.” I am afraid that they fact that the authors are not considering also controlling mosquito in standing water collections which are ideal habitats for anopheline and some culicine could affect the efficacy of the intervention”.

R: We understand reviewer’s concern. However, as we stated in “description of study site” section. It is quite rare to find surface standing water in the region where this study will be carried out. In general, the soil in Zanzibar archipelago is highly porous in a way that surface water only stands for few days. In fact, there is no surface river or lake in the archipelago. The hydrology has been dominated by a network of underground rivers running through as system of caves. The rivers can eventually re-emerge downstream, in some regions (particularly in some northern regions), to feed surface pools of water as natural springs. Larval habitats utilized by mosquitoes including Anopheles spp have been discarded water-holding containers and domestic utensils, plastic water bottles, plant pots, cans, and concrete water tanks.

The authors should consider adding a method of evaluation of the use of water dispensers by people living in the hotel to determine if the level of usage of these measures is similar between hotels because this could introduce some bias in the analysis.

R: Thank you for your suggestion. Details on the approach to evaluate the level of use of dispensers by guests have been included in methods, subsection entitled “evaluation of the use of water dispensers”. The description reads as follows:

Evaluation of the use and of water dispensers

A questionnaire survey to evaluate the level of acceptability and use of water dispensers will be carried out. Therefore, guests will be asked to answer to a questionnaire to determine the level of usage, preference, and satisfaction in using the water dispenser. The questionnaire is also intended to obtain suggestion for improvement. The questionnaire will ask question such as, how often do you use the water dispenser? At what times of the day do you typically use the water dispenser? Why do you choose to use the water dispenser? If you do not use the water dispenser, why not? Are you satisfied with the quality of the water from the dispenser? Are you satisfied with the location of the water dispensers in the hotel? What improvements would you suggest for the water dispenser service? Demographical information namely age, gender and provenance of guests will also be recorded.

The author indicated using BG sentinel traps for collecting adult mosquito I am afraid this method alone will be less efficient to collect adult mosquitoes I will advise them to use two or three different collection methods to increase chances of evaluating mosquito biting behaviour.

R: Thank you for point that out. Additional nocturnal collection to be applied concurrently is CO2-Baited light CDC light traps for sampling of non-Aedes mosquitoes. Moreover, collection of mosquitoes resting outdoors and indoors will be performed by using mechanical aspirator prokopack. These details have already been added for clarity. 

Reviewer #3: Unfortunately, just proposed work in the form of a protocol does not constitute as a manuscript in my opinion.

The introduction and the protocol are well written but without any solid data it does not qualify as a research paper.

R: Thank you for your comments. We presume that the reviewer may have forgotten that we have submitted a Study protocol paper, and not a Lab protocol or a Research paper. Study protocol does not include study results. Plos One web page clearly states "Study protocols describe detailed plans and proposals for research projects that have not yet generated results. They consist of a single article in PLOS ONE that can be referenced in future papers”.

Reviewer #4: Larval source reduction is very important mosquito control strategy particularly with the emergence of insecticide resistance among vector mosquitoes. Thus, making this protocol very important. However, detailed information on the study site is needed to understand the practicability of the protocol.

Are there any settlements around the study hotels?

Will there be an intervention in a settlement found within 300 m radius buffer zone.

R: These hotels are not surrounded by human settlements. As stated in the manuscript, these hotels are surrounded by either other hotels or open spaces covered by vegetations that have sometimes harboured discarded containers too. Therefore, also indicated in the manuscript, intervention such as container removal and vegetation clearing will also be performed within 300 metres radius at immediate hotel surrounding spaces to avoid formation of mosquito recruitment foci that may eventually confound the impact of the intervention being evaluated.

…“Since modelling abundance requires more information, it is proposed that a subset of species that were observed in at least ten sampling occasions be used to model the impact of the intervention on abundance” …..

Will Aedes species be the only target group to measure the impact of the intervention? If not, Anopheles and other mosquitoes have about 1-2 km flight range, how would mosquitoes that will come from breeding areas outside the hotel and the buffer zone be accounted for?

R: This paragraph has been rewritten for clarity. The goal is to target all vector species that may occur at hotel facilities, not just Aedes mosquito. We acknowledge that there may be species such as Anopheles gambiae that can actively engage in long-range dispersal and eventually negotiate the 300-metres buffer zone. According to our baseline survey, mosquito species capable of engaging in such a long-range dispersal or appetitive flight, such as Anopheles and Culex are quite rare at the study sites in comparison to species that have usually engaged in short-range dispersal flight such as Aedes and Eretmapodites.

Sanitation as a tool to control Aedes mosquitoes is not new. Compliance is always the problem. How will you ensure compliance among the hotel staffs and guest. For example, how would ensure that guest drink from the water dispenser but not bottled water?

The authors rightly indicated some of these questions as limitations but did not give detailed mitigation measures. However, without proper mitigation measures this protocol might not give the desirable results.

R: Thank you for your concern. As stated above, a qualitative questionnaire survey will be carried out to assess the of the use of the dispenser by guests. The questions to be asked to assess the use are described below, and they have been included in methods section entitle Evaluation of the use and of water dispensers

“The questionnaire will ask question such as, how often do you use the water dispenser? At what times of the day do you typically use the water dispenser? Why do you choose to use the water dispenser? If you do not use the water dispenser, why not? Are you satisfied with the quality of the water from the dispenser? Are you satisfied with the location of the water dispensers in the hotel? What improvements would you suggest for the water dispenser service? Demographical information namely age, gender and provenance of guests will also be recorded. The questionnaire is also intended to obtain suggestion for improvement.”

Regarding your concern about compliance in the non-use of plastic water bottle, the following statement has been included.

“To ensure compliance among guests and hotel staff. Moreover, hotel will be asked to implement educative and policy measures to encourage and ensure compliance, namely: placement of signage inside rooms and near the water dispenser displaying information on negative impact of plastic container to the environment.” 

This suggestion has been added to section entitled “Study design”.

---

## [Decision Letter · Decision Letter 1]

9 Nov 2023

A protocol for evaluating the entomological impact of larval source reduction on mosquito vectors at hotel compounds in Zanzibar

PONE-D-23-14190R1

Dear Dr. Kampango,

We’re pleased to inform you that your manuscript has been judged scientifically suitable for publication and will be formally accepted for publication once it meets all outstanding technical requirements.

Kind regards,

David Zadock Munisi, Ph.D

Academic Editor

PLOS ONE

Additional Editor Comments (optional):

Reviewers' comments:

Reviewer's Responses to Questions

**Comments to the Author**

1. Does the manuscript provide a valid rationale for the proposed study, with clearly identified and justified research questions?

Reviewer #4: Yes

2. Is the protocol technically sound and planned in a manner that will lead to a meaningful outcome and allow testing the stated hypotheses?

Reviewer #4: Yes

3. Is the methodology feasible and described in sufficient detail to allow the work to be replicable?

Reviewer #4: Yes

4. Have the authors described where all data underlying the findings will be made available when the study is complete?

Reviewer #4: Yes

5. Is the manuscript presented in an intelligible fashion and written in standard English?

Reviewer #4: Yes

6. Review Comments to the Author

You may also provide optional suggestions and comments to authors that they might find helpful in planning their study.

Reviewer #4: The authors have adequately addressed my concerns and questions. I do not have any further question.

7. PLOS authors have the option to publish the peer review history of their article (what does this mean?). If published, this will include your full peer review and any attached files.

Reviewer #4: **Yes: **Andreas A. Kudom

---

## [Editor Report · Acceptance letter]

14 Nov 2023

PONE-D-23-14190R1 

A protocol for evaluating the entomological impact of larval source reduction on mosquito vectors at hotel compounds in Zanzibar 

Dear Dr. Kampango:

I'm pleased to inform you that your manuscript has been deemed suitable for publication in PLOS ONE. Congratulations! Your manuscript is now with our production department. 

Kind regards, 

on behalf of

Dr. David Zadock Munisi 

Academic Editor

PLOS ONE